# Sensitive, Accurate and Rapid Detection of the Northern Root-Knot Nematode, *Meloidogyne hapla*, Using Recombinase Polymerase Amplification Assays

**DOI:** 10.3390/plants10020336

**Published:** 2021-02-10

**Authors:** Sergei A. Subbotin, Julie Burbridge

**Affiliations:** Plant Pest Diagnostic Center, California Department of Food and Agriculture, 3294 Meadowview Road, Sacramento, CA 95832, USA; burbridge.julie@gmail.com

**Keywords:** diagnostics, root-knot nematode, recombinase polymerase amplification

## Abstract

Rapid and reliable diagnostics of root-knot nematodes are critical for selections of effective control against these agricultural pests. In this study, recombinase polymerase amplification (RPA) assays were developed targeting the IGS rRNA gene of the northern root-knot nematode, *Meloidogyne hapla.* The RPA assays using TwistAmp^®^ Basic, TwistAmp^®^ exo and TwistAmp^®^ nfo kits (TwistDx, Cambridge, UK) allowed for the detection of *M. hapla* from crude extracts of females, eggs and juveniles without a DNA extraction step. The results of the RPA assays using real-time fluorescence detection (real-time RPA) in series of crude nematode extracts showed reliable detection after 13 min with a sensitivity of 1/100 of a second-stage juvenile and up to 1/1000 of a female in reaction tubes. The results of the RPA assays using lateral flow dipsticks (LF-RPA) showed reliable detection within 30 min with a sensitivity of 1/10 of a second-stage juvenile and 1/1000 of a female in reaction tubes. The RPA assay developed here is a successful tool for quick, accurate and sensitive diagnostics of *M. hapla*. The application of the LF-RPA assay has great potential for diagnosing infestation of this species in the lab, field or in areas with a minimal laboratory infrastructure.

## 1. Introduction

The northern root-knot nematode, *Meloidogyne hapla* is one of the four most common root-knot nematode species worldwide. This nematode is extremely polyphagous, attacking a wide variety of crops and weeds. *Meloidogyne hapla* causes important economic losses for several horticultural, vegetable and pasture crops, including carrots, lettuce, lucerne, onion, potato, rose, sugarbeet, strawberry, white clover and others [1,2].

Accurate and rapid identification of nematodes is essential for their control. It has been shown that sequences of nuclear ribosomal genes: 18S rRNA, ITS rRNA, the D2–D3 of 28S rRNA, IGS rRNA and mitochondrial genes: *COII*-16S rRNA fragment, *COI* and *COII* clearly differentiate *M. hapla* from all other root-knot nematodes [3]. Several specific primers have been designed for the diagnostics of this species using conventional PCR [4,5,6,7]. Several authors also developed a TaqMan real-time PCR assay with species-specific primers for the detection of *M. hapla* from root galls and soil samples [8,9,10,11,12]. Recently, Peng et al. [13] developed loop-mediated isothermal amplification methods (LAMP) combined with a Flinders Technology Associates card for the identification of *M. hapla*.

Recombinase polymerase amplification (RPA), an isothermal in vitro nucleic acid amplification technique, has recently appeared as a novel molecular technology for simple, robust, rapid, reliable, and low-resource diagnostics. RPA represents a hugely versatile alternative to PCR [14,15,16]. RPA uses a highly efficient displacement polymerase that amplifies a few copies of target nucleic acid in 20 min at a constant temperature (37–42 °C). It does so by utilizing three core enzymes: recombinase, single-stranded binding protein (SSB), and strand-displacing polymerase. The recombinase enzyme forms a complex with a primer to facilitate their binding to the targeted DNA template. Then, the SSB binds to the displaced strands of DNA and prevents the displacement of the recombinase–primer complex by branch migration. The strand-displacing polymerase then recognises the bound recombinase–primer complex and initiates DNA synthesis. Like PCR, RPA produces an amplicon constrained in size to the binding sites of the primers. The advantages of RPA include highly efficient and rapid amplification and a low constant operating temperature. RPA products can be detected by agarose gel electrophoresis or carried out by using fluorescent probes in real time (real-time RPA) or lateral flow strips (LF-RPA). RPA assays show high sensitivity and specificity for detecting various plant viruses, bacteria, fungi, vertebrate parasitic trematodes, nematodes and other organisms [17,18,19,20,21,22]. Real-time RPA detection assay of plant parasitic nematodes was first designed and published by Subbotin et al. [23] for *Meloidogyne enterolobii.* RPA assays were also developed for *Meloidogyne javanica*, *M. arenaria* and *M. incognita* [24,25] and *Bursaphelenchus xylophilus* [26,27]. Recently, Song et al. [28] described diagnostics of *Meloidogyne hapla* using RPA combined with a lateral flow dipstick assay, where species-specific primers and a probe were designed based on the effector gene *16D10* sequence. This LF-RPA assay allows detecting *M. hapla* from infested plant roots and soil samples and the entire detection process can be completed within 1.5 h.

In our study, we developed real-time RPA and LF-RPA assays for the detection of *Meloidogyne hapla* using crude nematode and infected plant root extracts, with results within 13–30 min. Species-specific primers and probes were designed based on the IGS ribosomal RNA gene sequence.

## 2. Results

### 2.1. RPA Primers and Probe Design

All available sequences of the IGS rRNA for *M. hapla* and other *Meloidogyne* were downloaded from the Genbank and aligned with ClustalX. Several regions with high sequence dissimilarity between *M. hapla* and other *Meloidogyne* were assessed and several species-specific *M. hapla* candidate primers sets and probes were manually designed. The Blastn search of these species-specific candidate primer sequences and probe sequences showed high similarity (100%) only with the IGS rRNA fragments of *M. hapla* deposited in the GenBank.

### 2.2. RPA Detection

Nine primer combination candidate sets were screened for the best performance under the same RPA conditions. The species-specific forward F3-IGS-Hapl and the species-specific reverse R3-IGS-Hapl primers were found to be optimal with clearly visible bands and had no cross-reactions with other root-knot nematodes (Table 1). The final sequences of primers and probes used for the assays are listed in Table 2 and are indicated in the IGS rRNA gene alignment in Figure 1. This primer set reliably and specifically amplified the target gene fragment, approximately 164 bp in length from the IGS region (Figure 2) and was also confirmed by a direct sequencing of the product. Additional non-specific weak additional bands having other sizes were observed sometimes in experiments with *M. hapla* as well as samples with other root-knot nematode species (data not shown).

### 2.3. Real-Time RPA Detection Assays

Using the results of nine experimental runs, which included positive and negative controls with water and non-target DNA, the threshold level for reliable *M. hapla* detection was established as equal to 8 cycles (~3 min) with a baseline of 250,000 (∆Rn) fluorescence using the TwistAmp^®^ exo kit on the Applied Biosystems™ QuantStudio™ 7 Flex Real-Time PCR System (Figure 3 and Figure 4A). Samples that produced an exponential amplification curve above the threshold were considered as positive for *M. hapla* and below the threshold were considered as negative. Detection of *M. hapla* was confirmed with all samples.

The RPA assay was tested for specificity using DNA extracted from several root-knot nematodes. These nematodes include: *Meloidogyne arenaria*, *M. baetica*, *M. christiei*, *M. enterelobii*, *M. floridensis*, *M. incognita*, *M. javanica*, *M. naasi* and *M. nataliei.* The RPA results using real-time fluorescent detection showed high specificity to *M. hapla* only and no cross-reactions were observed against other root-knot nematode species (Figure 3A).

The sensitivity assay was designed for evaluation of the detection limit. Variants with serial dilutions (1, 1/10, 1/100, 1/1000 and 1/10,000 per reaction tube) of crude nematode extractions were obtained from second-stage juveniles (J2s) or females without egg-masses. The reliable detection level of *M. hapla* was estimated at 1/100 of one J2 per a RPA reaction tube (Figure 3B). The detection level of *M. hapla* females varied among replicates and reached 1/100, 1/1000 and 1/10,000 of a female for three, two and one replicates, respectively (Figure 4A).

The detection of J2 for *M. hapla* was confirmed in the presence of background crude extracts from at least 20 non-target nematodes. No decrease in fluorescent signals was observed between the variant of 1 J2 without other nematodes and the variants with 1 J2 with 10 and 20 non-target nematodes (Figure 3C). Lowering in half, a single reaction assay volume showed a decrease in fluorescence signal and reaction rate (Figure 3C). These samples could be considered as positive with threshold level of 12 cycles (~6 min).

*Meloidogyne hapla* detection was also confirmed using extracts obtained from infected tomato and pepper plant roots containing females with egg-masses. Although most replicates from extracts obtained from infected plant roots containing old females without egg-masses gave strong signals, one replicate showed no fluorescence signal (Figure 3D).

### 2.4. LF-RPA Assay

Lateral flow detection of RPA products also showed specific and sensitive results. Positive test lines on the LF strips were observed for all *M. hapla* samples, whereas samples with other nematode species showed only a control line (Figure 5A). The detection of J2 for *M. hapla* was confirmed from extracts of infected pepper roots with *M. hapla* (Figure 5B) as well as in the presence of background crude extract from 10 to 20 non-target nematodes (Figure 5C). Lowering in half, a single reaction assay volume still detected *M. hapla* samples (Figure 5C). The results of RPA assays showed reliable detection with a sensitivity of 1/10 of a J2 (Figure 5D) and 1/1000 of a female (Figure 4B) in reaction tubes.

## 3. Discussion

Polymerase chain reaction is considered the gold standard of molecular detection, however, this method is only available in a laboratory with thermal cycling equipment. In this work, we have developed an affordable, simple, fast and sensitive real-time RPA and LF-RPA assays to detect *M. hapla* from nematode specimens extracted from plant and soil samples. An LF-RPA assay can be performed in a field condition without any special equipment or in areas with a minimal laboratory infrastructure.

Song et al. [28] described the LF-RPA diagnostic assay of *M. hapla* using species-specific primers and a probe designed using the effector gene *16D10* sequence. Authors stated that the entire detection process can be completed within 1.5 h, including 30–60 min for DNA extraction, 20 min for the RPA reaction, and 3–5 min for visual detection on the LF strips [28]. In our assay, the species-specific primers and probes were designed using the IGS rRNA gene sequence. The entire detection process for the LF-RPA assay can be completed within approximately 30 min, including 4 min for crude nematode extract preparation, 20 min (4 + 16) for the RPA reaction, 1 min for mixing and centrifugation of tubes, and 5 min for visual detection on the LF strips. The entire detection process for real-time RPA assay can be completed within approximately 13 min, including 4 min for crude nematode extract preparation, 8 min (5 + 3) for the RPA reaction with 1 min for mixing and centrifugation of tubes. This calculation does not include the time for preparation of the RPA reaction mixture.

In our study the IGS rRNA gene was selected for the RPA assays because of its high copy number and because previously published PCR studies have demonstrated its usefulness to distinguish *Meloidogyne* species [7,29,30]. Zhang et al. [16] noticed that different DNA targets are likely to have extremely different amplification efficiencies, even sharing a series of common characteristics including GC content, primer melting temperature and RPA product length. These authors also concluded that primers are the most important determinant for RPA performance including sensitivity, specificity and reaction rate. Although amplicons obtained from fragments of effector gene *16D10* and the IGS rRNA gene are comparable (148 vs. 164 bp) in a length, it seems that the RPA reaction rate is higher for the IGS rRNA gene than the effector gene fragment. The RPA assays developed based on the IGS rRNA gene are more sensitive for detection than assays based on the effector gene fragment.

Song et al. [28] reported about 1/1000 female (after DNA extraction with proteinase K) as the detection limit of the LF-RPA assay, whereas in our LF-RPA assay, the detection limit can reach up to 1/10,000 (without a special DNA extraction step). Our RPA assays for *M. hapla* also showed higher amplification rates compared with similar assays developed for *M. enterolobii*, in which species-specific primers were also designed based on the IGS rRNA gene [23]. The threshold level for the reliable *M. enterolobii* detection was established as equal to 30 cycles (=10 min) and a baseline of 500,000 (ΔRn) fluorescence level with the TwistAmp^®^ exo kit and Applied Biosystems™ QuantStudio™ Flex Real-Time PCR System, whereas in our present study for *M. hapla*, the threshold level was estimated equal to 8 cycles (~3 min) and a baseline of 250,000.

The real-time RPA and LF-RPA assays developed in our study allowed the detection of J2, young females without and with eggs-masses. The old and dead females without body contents might not be always detectable using this method. The results of our study estimated that the reliable detection of RPA assays using real-time fluorescence were 1/100 of J2 or female and using lateral flow dipsticks were 1/10 of a J2 and 1/1000 of a female. However, in some replicates the detection limit can reach up to 1/10,000 of a female in reaction tube. Reproducibility of the assays in low concentrations of nematode extracts, extracts from old females or directly from soil samples should be carefully tested further to understand factors, which might have an influence on the performing stability of RPA reactions. RPA diagnostics of root-knot nematodes has several other important advantages over PCR methods. The first advantage is that crude nematode extracts or crude extracts from nematode-infected tomato and pepper plant tissues can be directly used for RPA assays, whereas PCR assays require a DNA extraction step with special treatment of these extracts before use. The second advantage of RPA assays is that results are available in 8–20 min, whereas the results of PCR assays can be observed in 1.5–3 h. The third advantage is higher sensitivity levels of RPA detection over PCR methods; the RPA assay is 10 or 100 times more sensitive than PCR.

However, the application of RPA assays for nematode diagnostics may still face several problems, with cost being a major consideration. Factors affecting the expense of assays depends on the pest, reagent costs, requirement for equipment, infrastructure capacity, employee wages and numbers of samples for testing, among others. The RPA reagents and kits are presently manufactured by only one company, TwistDx, Inc., making the cost of the RPA assay relatively higher than other PCR assays. Reagent costs for RPA assays are currently in range of USD 4.3–5.5 per sample [15] which is higher than for conventional and real-time PCR. Lillis et al. [31] showed that lowering the assay volume from 50 μL, which is the recommended in the manufacturer’s protocol, to 5 μL showed similar sensitivity. Our experiments also revealed acceptable diagnostic performance when reducing the reaction volume by half. This approach could be used in cases of resource limitations. It has been noticed that special attention should be paid to the potential of cross-contamination due to the high sensitivity of this reaction. The requirement for use of clean gloves, tubes, and pipets should be carefully considered during the use of RPA assays in a field condition. Thus, RPA has the potential to be a promising alternative to PCR and other methods for rapid detection of nematodes. This assay requires minimal sample preparation, making it ideal for use in the lab, the field, or minimal laboratory infrastructure.

## 4. Materials and Methods

### 4.1. Nematode Samples

Five isolates of *Meloidogyne hapla* were obtained for RPA assay development. Second-stage juveniles (J2s) and females were extracted from the root or soil samples. The D2–D3 expansion segments of the 28S rRNA gene were sequenced from each isolate to confirm its identity. DNA of several root-knot nematodes, *M. arenaria*, *M. baetica*, *M. christiei*, *M. enterelobii*, *M. floridensis*, *M. incognita*, *M. javanica*, *M. naasi* and *M. nataliei* were also used in specificity experiments (Table 1). These species were also identified by molecular methods. Free-living and plant parasitic nematodes from several field samples collected in California were extracted using the centrifugal flotation method and their extracts were used as background non-target DNA.

### 4.2. Nematode Extracts

Second-stage juveniles or females of *M. hapla* were placed in distilled water on a microscope slide. The nematodes were cut using a dental needle under a stereo microscope and put into a 0.2 mL PCR tube with a total volume of 10 μL. This stock crude extract was used to make a series of dilutions sequentially: 1:2, 1:4, 1:8, 1:10, 1:16, 1:100, 1:1000 and 1:10,000 in water. Several extracts were prepared: (i) J2s; (ii) J2s with other non-target nematodes; (iii) female and (iv) plant gall tissue with one or more females and egg-masses. Crude extract of plant gall tissue with nematodes and crude extract of several hundred non-target nematodes soil free-living and plant parasitic nematodes were also obtained by crushing the samples on a microscope slide using a plastic pipe tip or dental needle.

### 4.3. RPA Primer Design and Testing

A total of three forward and three reverse RPA primers specific to *M. hapla* were manually designed based on species sequence polymorphisms in the IGS rRNA gene. Primers were synthesized by Integrated DNA Technologies, Inc. (Redwood City, CA, USA). Nine primer sets were screened in different combinations using the TwistAmp^®^ Basic kit (TwistDx, Cambridge, UK). Reactions were prepared according to the manufacturer’s instructions. The lyophilized reaction pellets were suspended in 29.5 μL of the rehydration buffer, 2.4 μL of each forward and reverse primers (10 μM) (Table 2), 1 μL of the DNA template or nematode extract and 12.2 μL of distilled water. For each sample, 2.5 μL of 280 mM magnesium acetate was added to the lid of the tube and the lids were closed carefully. The tubes were inverted 10–15 times and briefly centrifuged to initiate reactions simultaneously. Tubes were incubated at 39 °C (4 min) in a MyBlock Mini Dry Bath (Benchmark Scientific, Edison, NJ, USA) and then they were inverted 10–15 times, briefly centrifuged and returned to the incubator block (39 °C) for 20 min. Sample tubes were then placed in a freezer to stop the reaction. Amplification products were purified with a QIAquick PCR Purification Kit (Qiagen, Valencia, CA, USA). Five μL of purified product were run in a 1% TAE (Tris-acetate-EDTA)-buffered agarose gel (100 V, 60 min) and visualized with Gel Green stain. Amplification products were directly sequenced by Genewiz (San Francisco, CA, USA) using amplification primers.

### 4.4. Real-Time RPA Assay

Two TwistAmp^®^ exo probes were designed according to the manufacturer’s instructions and were synthesized by Biosearch Technologies, Inc. (Petaluma, CA, USA). Two probes were tested and only one probe (Probe-hapla-exo1) was selected for the assay (Table 2) based on best amplification performance. The real time detection of RPA assay products was accomplished using the TwistAmp^®^ exo kit (TwistDx, Cambridge, UK). The lyophilized reaction pellets were suspended in 29.5 μL of the rehydration buffer, 2.1 μL of each forward and reverse primers (10 μM) (Table 2), 0.6 μL of the probe (10 μM), 1 μL of the DNA template or nematode extract and 12.2 μL of distilled water. Magnesium acetate in a volume of 2.5 μL was added to the lid of each tube, the lids were carefully closed, tubes were inverted 10–15 times and briefly centrifuged. The reaction tubes were incubated at 39 °C for 5 min, then inverted 10–15 times to mix, and briefly centrifuged. The tubes were immediately placed in Applied Biosystems™ QuantStudio™ 7 Flex Real-Time PCR System to incubate at 39 °C for 15 min. The fluorescence signal was monitored in real time and measured every 20 s (cycle) using the fluorophore (FAM) channel. A positive control using *M. hapla* extract (one J2 per reaction tube) and negative control without any nematode DNA were included in each run. Two or three replicates of each variant across several runs were performed for sensitivity and specificity experiments.

### 4.5. LF-RPA Assay

Two TwistAmp^®^ nfo probes were designed according to the manufacturer’s instructions and tested in the same conditions. Only one probe (Probe-hapla-nfo1) was selected for the assay based on the best visualization results. The LF-RPA assay products were accomplished using the TwistAmp^®^ nfo kit (TwistDx, Cambridge, UK). The reaction mixture for each RPA assay was prepared according to the manufacturer’s instructions: the lyophilized reaction pellets were suspended in 29.5 μL of the rehydration buffer, 2.1 μL of each forward and reverse primers (10 μM), 0.6 μL of the probe (10 μM), 1 μL of the DNA template or nematode extract and 12.2 μL of distilled water. Magnesium acetate in a volume of 2.5 μL was added to the lid of each tube, the lids were carefully closed, and the tubes were inverted 10–15 times and briefly centrifuged. The reaction tubes were incubated at 39 °C for 4 min, then inverted 10–15 times to mix, briefly centrifuged and returned to the incubator block at 39 °C for 16 min. The tubes were placed in the freezer to stop the reaction. For visual analysis with Milenia^®^ Genline Hybridetect-1 strips (Milenia Biotec GmbH, Giessen, Germany), testing solution containing 48 μL of HybriDetect assay buffer and 12 μL of the sample RPA product was prepared in a 0.5 mL PCR tube. Ten μL of the testing solution was placed directly onto the sample area of the dipstick. Dipsticks were placed upright into 100 μL of the assay buffer and visual results were observed within 5 min. The amplification product was indicated by the development of a colored test line, and/or a separate control line to confirm that the system worked properly (Figure 4B and Figure 5). Two or three replicates of each variant were performed for sensitivity and specificity experiments.

## Figures and Tables

**Figure 1 plants-10-00336-f001:**
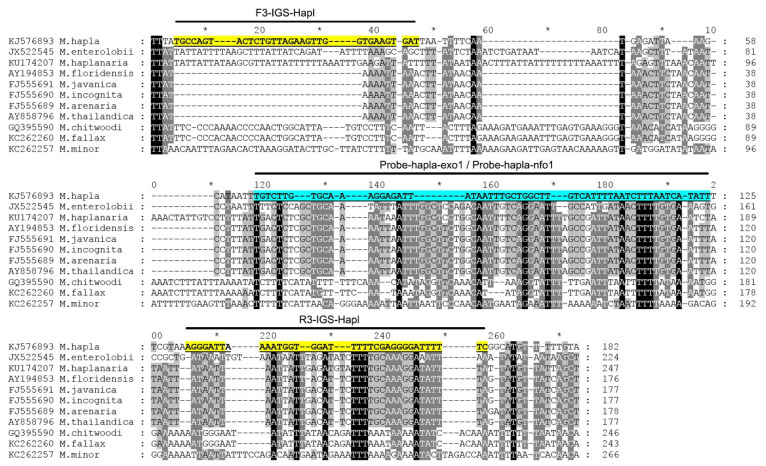
The fragment of alignment of the IGS rRNA gene sequences for several root-knot nematodes, *Meloidogyne*, with the positions of recombinase polymerase amplification (RPA) primers and probes used in the present study.

**Figure 2 plants-10-00336-f002:**
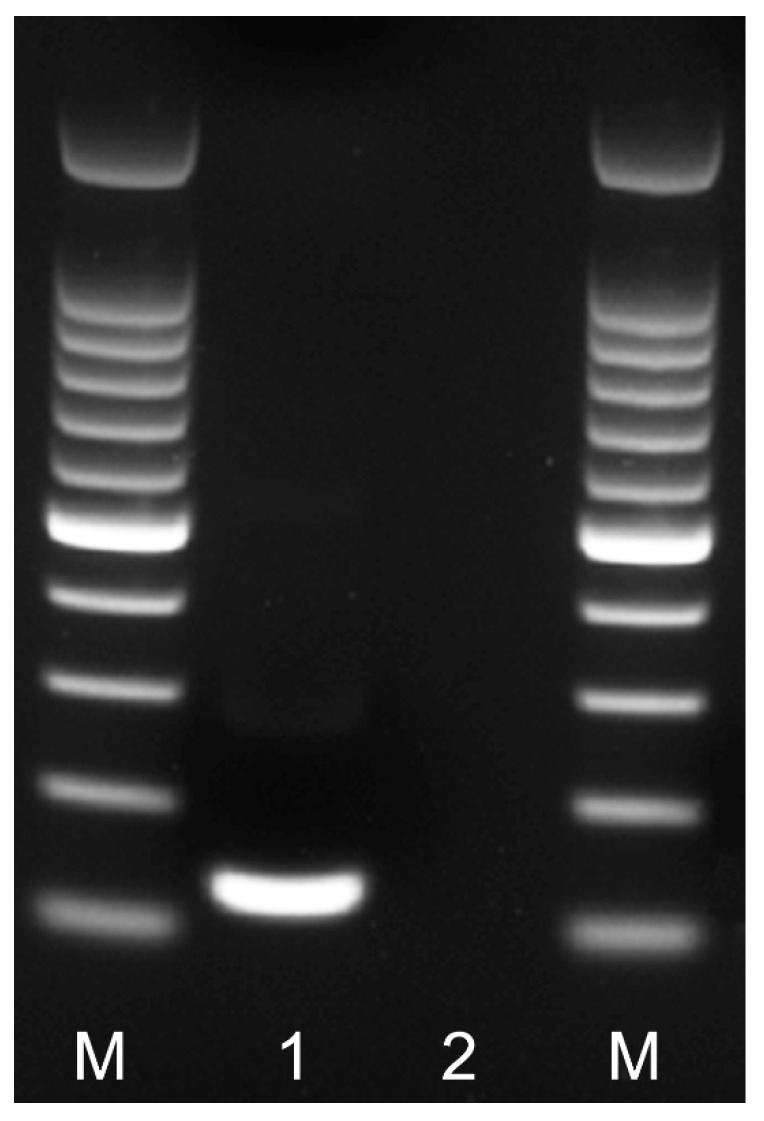
RPA amplicon of the partial IGS rRNA gene on agarose gel. Lanes: M: 100 bp DNA marker (Promega, Madison, WI, USA); 1: RPA amplicon obtained after 24 min at 39 °C with F3-IGS-Hapl and R3-IGS-Hapl primers using TwistAmp^®^ Basic kit; 2: negative control.

**Figure 3 plants-10-00336-f003:**
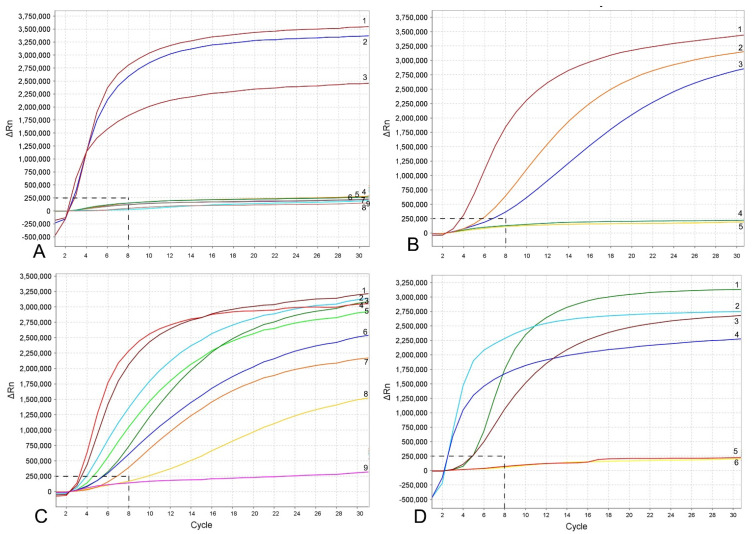
RPA assays using real-time fluorescent detection with examples of amplification plots. (**A**) Specificity assay with DNA samples of *Meloidogyne* spp. and crude second-stage juvenile (J2) extracts of *M. hapla.* Line: 1: *M. hapla* (CD2461); 2: *M. hapla* (VW9); 3: *M. hapla* (C44); 4: *M. incognita* (CD3038); 5: *M. arenaria* (CD3100); 6: *Meloidogyne naasi* (CD3381); 7: *M. javanica* (isolate 40); 8 and 9: negative control; (**B**) sensitivity assay with a dilution series of a crude J2 extract of *M. hapla*, line: 1: 1 J2 per tube; 2: 1/10 J2 per tube; 3: 1/100 J2 per tube; 4: 1/1000 J2 per tube; 5: negative control; (**C**) crude extract of *M. hapla* with or without crude extracts of non-target nematodes. Line: 1 and 4: 1 J2 per tube; 2, 5L 1 J2 with 20 non-target nematodes per tube; 3 and 6: 1 J2 with 10 non-target nematodes per tube; 7 and 8: 1 J2 per tube containing half of a reaction mixture; 9: negative control; (**D**) testing of crude extracts of *M. hapla.* Line: 1: 1 J2 per tube; 2 and 4: extracts from infected plant roots containing females with egg-masses; 3 and 6: extracts from infected plant roots containing old females without egg-masses; 5: negative control. The vertical line on a graph: fluorescence ∆Rn. ∆Rn is calculated at each cycle as DRn (cycle) = Rn (cycle)—Rn (baseline), where Rn = normalized reporter. The horizontal line on a graph: cycles, each cycle = 20 s.

**Figure 4 plants-10-00336-f004:**
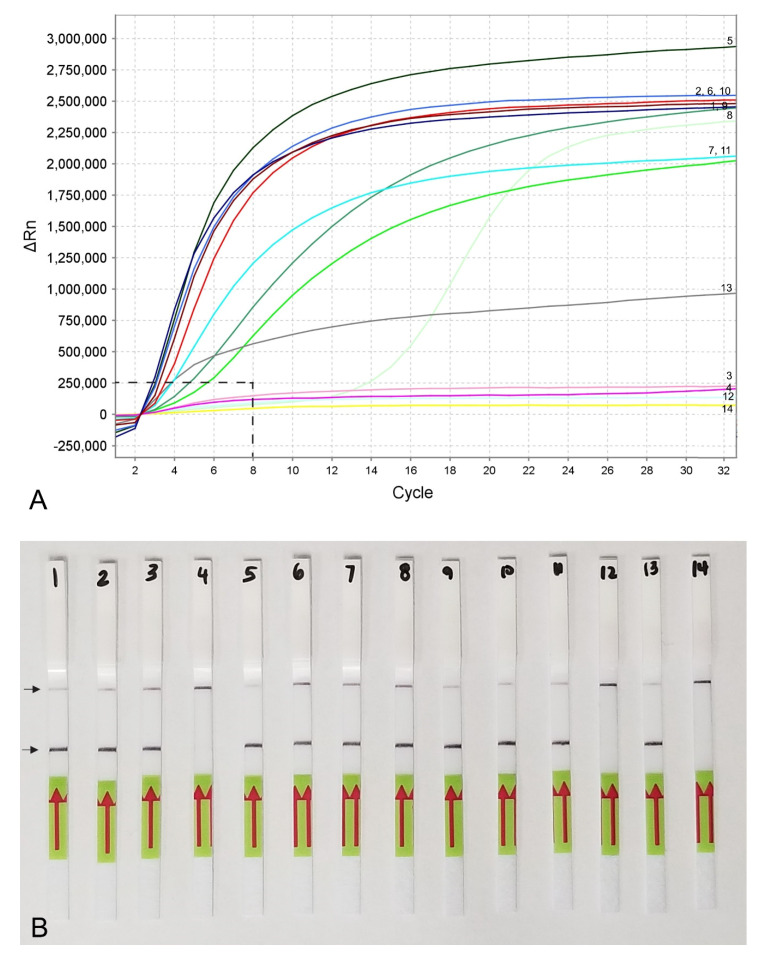
RPA sensitivity assays using (**A**) real-time fluorescent detection and (**B**) lateral flow strips. A dilution series of three crude young females (without egg-masses) extracts of *M. hapla*. Line (Strip): 1, 5, 9: 1/10 female per tube; 2, 6, 10: 1/100 female per tube; 3, 7, 11: 1/1000 female per tube; 4, 8, 12: 1/10,000 female per tube; 13: positive control; 14: negative control. Control (upper) and test (lower) lines are indicated by arrows.

**Figure 5 plants-10-00336-f005:**
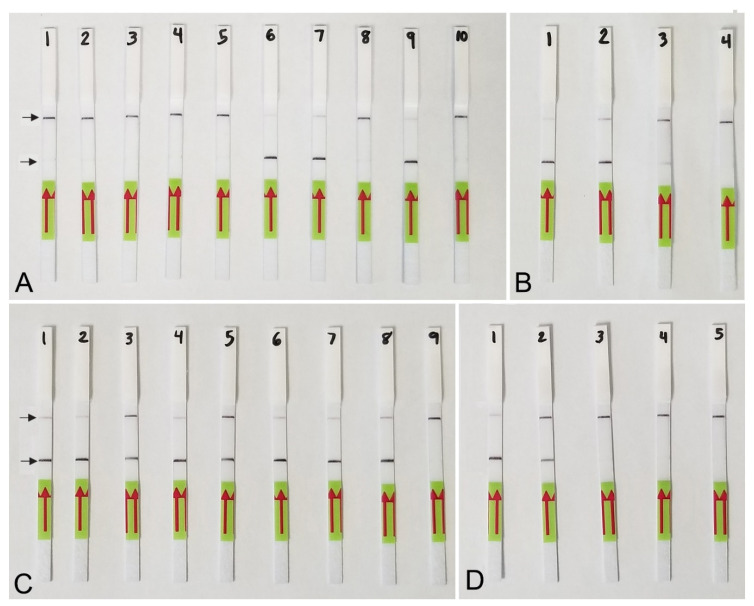
Lateral flow recombinase polymerase amplification (LF-RPA) assay with examples of lateral flow strips. (**A**) Specificity assay with DNA samples of *Meloidogyne* spp. and crude J2 extracts of *M. hapla.* Strip: 1 and 2: *Meloidogyne* sp.1 (CD3380); 3: *M. naasi* (CD3381); 4: *M. baetica* (CD3382); 5: *Meloidogyne* sp.2 (CD3383); 6 and 7: *M. hapla*, (CD3384); 8: *M. arenaria* (CD33093); 9: *M. hapla* (C44); 10: negative control; (**B**) crude extracts of *M. hapla.* Strip: 1 and 2: extracts from infected tomato roots containing plant materials and females with egg-masses; 3: 1 j2 per tube; 4: negative control; (**C**) testing of crude extract of *M. hapla* with or without crude extracts of non-target nematodes. Strip: 1 and 2: 1 J2 per tube; 3 and 4: 1 J2 with 10 non-target nematodes per tube; 5 and 6: 1 J2 with 20 non-target nematodes per tube; 7 and 8: 1 J2 per tube containing half of a reaction mixture; 9: negative control; (**D**) sensitivity assay with a dilution series of a crude j2 extract of *M. hapla*, Strip: 1: 1 J2 per tube; 2: 1/10 J2 per tube; 3: 1/100 J2 per tube; 4: 1/1000 J2 per tube; 5: negative control (upper) and test (lower) lines are indicated by arrows.

**Table 1 plants-10-00336-t001:** Samples of *Meloidogyne hapla* and other root-knot nematodes tested in the present study.

Species	Location	Plant	Sample Code	Source
*M. hapla*	USA, California	Tomato	VW9	V. Williamson
*M. hapla*	USA, California	Tomato	C44	V. Williamson
*M. hapla*	Moldova, Tiraspol	Sweet pepper	CD3384	V. N. Chizhov
*M. hapla*	USA, Balm, Florida	Strawberry	CD2461	R.N. Inserra
*M. hapla*	USA, Michigan, Van Buren County	Grapevine	CD3385e	S. Álvarez-Ortega
*M. arenaria*	USA, Florida	Unknown	CD3093	J.A. Brito
*M. arenaria*	USA, Florida	Unknown	CD3100	J.A. Brito
*M. baetica*	Spain	Olive	CD3382	P. Castillo
*M. christiei*	USA Florida,	Turkey oak	CD1471	J.A. Brito
*M. enterelobii*	USA, UCR collection	Tomato	CD3386	P. Roberts
*M. floridensis*	USA, California, Kern county	Grapevine	CD3324	S.A. Subbotin
*M. incognita*	USA, Florida	Tomato	CD3038	J.A. Brito
*M. javanica*	USA, Florida	Tomato	CD3050	J.A. Brito
*M. javanica*	USA, UCR collection	Tomato	Isolate 40	P. Roberts
*M. naasi*	Germany	Grasses	CD3381	D. Sturhan
*M. naasi*	USA, California	Grasses	CD2158	S.A. Subbotin
*M. nataliei*	USA, Michigan, Van Buren County	Grapevine	CD3385a, b, c	S. Álvarez-Ortega
*Meloidogyne* sp.1	Germany	Grasses	CD3380	D. Sturhan
*Meloidogyne* sp.2	Russia	Unknown	CD3383	V. N. Chizhov

**Table 2 plants-10-00336-t002:** RPA primers and probe for amplification of *Meloidogyne hapla* DNA.

Primer or Probe	Sequence (5′–3′)
F3-IGS-Hapl	TGC CAG TAC TCT GTT AGA AGT TGG TGA AGT GAT
R3-IGS-Hapl	GAA AAA TCC CCT CGA AAA ATC CAC CAT TTT AAT CCC T
R3-IGS-Hapl-biotin	[Biotin] GAA AAA TCC CCT CGA AAA ATC CAC CAT TTT AAT CCC T
Probe-hapla-exo1	T GTC TTG TGC AAA GGA GAT TAT AAT TTG CTG GCT [FAM-dT] GT [THF] AT [BHQ1-dT] TTA ATC TTT AAT CAT ATT[C3-spacer] *
Probe-hapla-nfo1	[FAM] T GTC TTG TGC AAA GGA GAT TAT AAT TTG CTG GCT TGT [THF] ATT TTA ATC TTT AAT CAT ATT[C3-spacer] *

* FAM—fluorophore, THF—tetrahydrofuran, BHQ—quencher, C3—spacer block.

## Data Availability

The datasets generated during and/or analyzed during the currentstudy are available from the corresponding author on reasonable request.

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
