# Peer review of "Sensitive, Accurate and Rapid Detection of the Northern Root-Knot Nematode, Meloidogyne hapla, Using Recombinase Polymerase Amplification Assays"

_plants, 2021, doi:10.3390/plants10020336_

Round 1

Reviewer 1 Report

Minor Revision.

A brief paper  with excellent English and a well-designed structure. Two variants of the latest express technology (Real-time RPA and LF-RPA ) have been developed for the identification of a widespread dangerous nematode  pathogen RKN Meloidogyne hapla  that  infected wide range of the  vegetable and forage pasture crops.

The novelty of the method lies in the use of the IGS rRNA sequence (selected from the Genbank with special analytical search) rather than the effector gene and in the detection speed of 13 min for LF-RPA to 30 min for Real-time RPA. The use of half of the reaction volume maintaining detection sensitivity has been proposed to reduce the cost of the assay. The sensitivity of the technology is as follows: 1/10 of an infective nematode juvenile and 1/1,000 of a female and even the use of the crude living worms or their eggs in a mixture with non-target nematodes, the tech looks very promising for growers.

The limitations of the method in the field are obvious: there is a centrifugation phase and a procedure for preparing the reaction reagent mix, which requires a minimal laboratory infrastructure. A more detailed description of the procedure for field conditions (tent, auto) is desirable.

Author Response

Point 1: The limitations of the method in the field are obvious: there is a centrifugation phase and a procedure for preparing the reaction reagent mix, which requires a minimal laboratory infrastructure. A more detailed description of the procedure for field conditions (tent, auto) is desirable.

Response 1: We made correction and added - ”areas with an minimal laboratory infrastructure”. In this manuscript we do not discuss some problems of mobile field pest diagnostic lab. We discussed the application in our previous publication: Subbotin, S.A. Recombinase polymerase amplification assay for rapid detection of the root-knot nematode Meloidogyne enterolobii. Nematology 2019, 21, 243-251.” RPA has good flexibility to be adapted to various detection systems. For example, instead of using a cost-effective heat block, portable fluorescence readers (ESEQuant, Qiagen; Genie, OptiGene; T8-ISO, TwistDX; and others) can be used, which are simpler and less expensive than a real-time PCR machine and can be run on battery power for use in the field (James & Macdonald, 2015; Daher et al., 2016)." The information on field pest diagnostic lab could be found in the website of Fera Ltd. https://www.fera.co.uk/our-science/active-r-and-d/in-field-diagnostics as for portable centrifuge https://www.youtube.com/watch?v=-wS0kbJHYlc

Reviewer 2 Report

The paer by Subbotin & Burbridge on detection of Meloidogyne hapla using RPA assays, it sis of nematological and phytopathological interest. The molecular assays were carried out following the highest standards in nematology and molecular biology. The manuscript is very well written and deserves publication in Plants. I only have some minor suggestions. Great job.

L77-78, remove - after Table 2. and do not underline Figure 1.

L116, replace a J2 by one J2.

Author Response

Response to Reviewer 2 Comments

Point 1: L77-78, remove - after Table 2. and do not underline Figure 1.

Response 1. Corrected.

Point 2: L116, replace a J2 by one J2.

Response 2. Corrected.

Reviewer 3 Report

This paper uses recombinase polymerase amplification assays to detect Meloidogyne hapla in crude extracts of J2s and females without a DNA extraction step. The results are clearly presented and I only have a few comments/queries.

line 77: my copy did not have Table 2. Table 1 is only mentioned in Methods, line208. A reference could be added around line 112.

Figure 3B: lines 105/6 and 142: why is there a large difference between 3 and 6?

Figure 5A: why such a large difference between 8 and 12 and between 7,11 and 3? This suggests the experiments should have been done with larger n numbers and stats could have been applied. Did the authors apply stats in their experiments, if so, please include them? In figure 5 are these young females with eggs?

Figure 4 and 5B: not clear what this figure shows, please explain it more clearly in the legend for the reader who may not be familiar with the method. There is an explanation in 2.4 lines 144/9 but it is not clear when viewing the figure what one is looking for. 

Please give the name of the infected plant roots used in this study. 

Author Response

Response to Reviewer 3 Comments

Point 1: line 77: my copy did not have Table 2. Table 1 is only mentioned in Methods, line208. A reference could be added around line 112.

Response 1: Table 2 is inserted in the text

Point 2: Figure 3B: lines 105/6 and 142: why is there a large difference between 3 and 6?

Response 2: In Figure 3D, sample 3 showed a detection, but sample 6 did not. We mentioned this result in line 148/149 and also in new paragraph in Discussion (see lines 195-197).

Point 3: Figure 5A: why such a large difference between 8 and 12 and between 7,11 and 3? This suggests the experiments should have been done with larger n numbers and stats could have been applied. Did the authors apply stats in their experiments, if so, please include them? In figure 5 are these young females with eggs?

Response 3.: These results are mentioned in in lines 121-122. We discussed this issue in new in new paragraph in Discussion (see lines 195-197). We also pointed that “Reproducibility of the assays in low concentrations of nematode extracts, extracts from old females or directly from soil samples should be carefully further tested to understand factors, which might influent on performing stability of RPA reaction.” To resolve situation with the sensitivity we consider “reliable detection”, when positive signal is observed in all replicates. In figure 5 we showed the results of extract from young females without eggs-sac. Appropriate correction in the Legend is made.

Point 4: Figure 4 and 5B: not clear what this figure shows, please explain it more clearly in the legend for the reader who may not be familiar with the method. There is an explanation in 2.4 lines 144/9 but it is not clear when viewing the figure what one is looking for.

Response 4.: Description of LF-RPA assay is given in section 4.5. We also introduced a reference to Figure 5: “The amplification product was indicated by the development of a colored test line, and a separate control line confirms that the system did work properly (Figs 4 and 5B)”.

Point 5: Please give the name of the infected plant roots used in this study.

Response 5.: Names of plants are given in appropriate sections.

Reviewer 4 Report

This paper describes the development of a PCR-based rapid detection assay for northern root-knot nematodes (RKNs). The purpose of the work is clear. The manuscript is well written. I have only a few concerns. Can the assay detect M. hapla in plant roots or soil samples? It is important to demonstrate that the assay can be used to detect RKNs in plant roots and soil samples. There are minor typographical errors throught the text.

Author Response

Point 1: There are minor typographical errors through the text.

Response 1: The manuscript test was again checked for typographical errors.

Reviewer 5 Report

Review for manuscript, “Sensitive, accurate and rapid detection of the northern root-know nematode, Meloidogyne haply, using recombinase polymerase amplification assays”, by Subbotin and Burbridge.

This manuscript describes a series of experiments that establish the utility of the recombinase polymerase amplification (RPA) assay, in real-time fluorescence and lateral-flow versions. Both types of RPA assay are important alternatives to standard PCR assays due to their potential for use in field applications. The current work explores in particular the rapid, sensitive, and specific, detection of an important agricultural pest species, M. hapla, a root-knot nematode.

In general, the research described in this manuscript has considerable value, and the assays proposed by the authors appear to be validated and useful. The manuscript itself is lacking in certain key elements and, therefore, needs to be revised in order to present complete evidence of the authors’ results.

Major Revisions:

  1. Line 77 mentions a Table 2, which does not appear in the manuscript. Table 2 should be presented between sections 2.2 and 2.3.
  2. Figure 3, Panel D:  Lines 3 and 6 are both extracts from infected plant roots containing old females without egg masses. Is there an explanation why the two lines are so divergent? This is mentioned later, on line 142, but not adequately explained. Furthermore, this should be discussed in the Discussion section.
  3. Figure 5: Line/strip 13:  What exactly does the positive control consist of? Why is this the only experiment that includes a positive control? I might assume that negative controls consisted of dH2O or buffer, but maybe they contained nematode tissue expected to test negative, or something else. Be more specific.
  4. Figure 5: Lines/strips 3, 7, 11 are triplicate assays of 1/1000 female per tube; the strips appear to be pretty similar, with 3 & 7 looking the most similar, and 11 a bit different; yet in the fluorescence assay, line 3 is closer to the negative control? This doesn’t make sense to me.
  5. Discussion:  The discussion section focusses on assay speed and cost, but does not reiterate the most significant discoveries in this work, nor does it discuss reasons for apparent ambiguities in the results (i.e., points noted above, and below under Minor Revisions.
  6. Materials and Methods: Line 214: What solution were the serial dilutions done in? dH2O, buffer? If buffer, what did it consist of? How were serial dilutions made? Was the tissue crushed, mixed - how? The non-target nematodes were homogenized. Was this the same procedure?  Line 223: What was the rehydration buffer?

Minor Revisions:

  1. Line 28: The authors state that M. hapla is one of the most common species. I believe they mean to say that it is one of the most common root-knot nematode species, or something of the like.
  2. Is there a reference for Flinders Technology Associates card?
  3. Figures 4 and 5 are out of order with respect to their mention in the Results. Also, mention of panels within Fig. 4 are out of order with respect to mention in the text on lines 146-149.
  4. Figure 4 figure legend: For clarity, the legend should state which probe was used in the experiment. The authors are advised that figures should stand on their own, and not require the reader to search text to find this kind of information.
  5. There are experimental duplicates presented in figures for some tests, but not for others. This is inconsistent, and raises the question why the authors either did not perform duplicates, or opted to not present all duplicate tests. For example, Figure 4, Panel A, strip 6 and 7 appear to be experimental duplicates, whereas other tests were not presented in duplicate. Or, are these strips different?
  6. Figure 4, Panel B: It appears that strip 3 (1 j2 per tube) gives a negative result (looks almost the same as the negative control). Why is this?
  7. There is inconsistency in the use of capitalization for “j2” versus “J2”.
  8. Figure 4, Panels B and C:  Should Panel B strip 3 and Panel C strip 1 not be producing essentially the same signal? They appear to be the same test.
  9. Materials and Methods: Line 209:  Should be, California, USA.

Author Response

Major Revisions:

Point 1: Line 77 mentions a Table 2, which does not appear in the manuscript. Table 2 should be presented between sections 2.2 and 2.3.

Response 1: Table 2 is inserted in the manuscript text.

Point 2. Figure 3, Panel D: Lines 3 and 6 are both extracts from infected plant roots containing old females without egg masses. Is there an explanation why the two lines are so divergent? This is mentioned later, on line 142, but not adequately explained. Furthermore, this should be discussed in the Discussion section.

Response 2: We mentioned this results in line 149 and discussed this issue in new paragraph of Discussion (line 195-201). In this case we have an issue of reproducibility of the assays in low concentrations of nematode extracts, because old females might not contain any nematode tissue and DNA.

Point 3. Figure 5: Line/strip 13: What exactly does the positive control consist of? Why is this the only experiment that includes a positive control? I might assume that negative controls consisted of dH2O or buffer, but maybe they contained nematode tissue expected to test negative, or something else. Be more specific.

Response 3. Corrected in Materials and Methods. Line 267

Point 4. Figure 5: Lines/strips 3, 7, 11 are triplicate assays of 1/1000 female per tube; the strips appear to be pretty similar, with 3 & 7 looking the most similar, and 11 a bit different; yet in the fluorescence assay, line 3 is closer to the negative control? This doesn’t make sense to me.

Response 4. All test lines (low) in 3, 7 and 11 strips have same intensively, only control line is not so intensive in 11 strip. Unfortunately, I cannot explain it. Please, note that this test showed the results of reaction with low extract nematode concentration (1/1,000 female per tube) and RPA reaction seems to be not be stable in such concentration. Please, also note that in the normal diagnostic condition, a researcher  deals with and use a whole single female or whole single J2 but not its small portion.

Point 5: Discussion: The discussion section focusses on assay speed and cost, but does not reiterate the most significant discoveries in this work, nor does it discuss reasons for apparent ambiguities in the results (i.e., points noted above, and below under Minor Revisions.

Response 5: New Paragraph is introduced in Discussion section. See lines 195-201

Point 6. Materials and Methods: Line 214: What solution were the serial dilutions done in? dH2O, buffer? If buffer, what did it consist of? How were serial dilutions made? Was the tissue crushed, mixed - how? The non-target nematodes were homogenized. Was this the same procedure? Line 223: What was the rehydration buffer?

Response 6: Corrected. The serial dilutions were done in a water.

Corrected: a series of dilutions were made sequentially

Rehydration buffer is provided by company.

Minor Revisions:

Point 1. Line 28: The authors state that M. hapla is one of the most common species. I believe they mean to say that it is one of the most common root-knot nematode species, or something of the like.

Response 1: Corrected

Point 2. Is there a reference for Flinders Technology Associates card?

Response 2: Reference - Peng et al. [13]

Point 3. Figures 4 and 5 are out of order with respect to their mention in the Results. Also, mention of panels within Fig. 4 are out of order with respect to mention in the text on lines 146-149.

Response 3. It is essential for reader to keep Real-time and LF-RPA results are in same Figure plate.

Point 4. Figure 4 figure legend: For clarity, the legend should state which probe was used in the experiment. The authors are advised that figures should stand on their own, and not require the reader to search text to find this kind of information.

Response 4. Added in Materials and Methods

Point 5. There are experimental duplicates presented in figures for some tests, but not for others. This is inconsistent, and raises the question why the authors either did not perform duplicates, or opted to not present all duplicate tests. For example, Figure 4, Panel A, strip 6 and 7 appear to be experimental duplicates, whereas other tests were not presented in duplicate. Or, are these strips different?

Response 5. All experiments were run in several replicates. In the Legends we indicated that examples of amplification plots and strips are presented.

Point 6. Figure 4, Panel B: It appears that strip 3 (1 j2 per tube) gives a negative result (looks almost the same as the negative control). Why is this?

Response 6. Strip 3 (1 j2 per tube) gives a week band and it is positive result.

Point 7. There is inconsistency in the use of capitalization for “j2” versus “J2”.

Response 7. Corrected.

Point 8. Figure 4, Panels B and C: Should Panel B strip 3 and Panel C strip 1 not be producing essentially the same signal? They appear to be the same test.

Response 8. Intensively of control band staining are varied among replicates.

Point 9. Materials and Methods: Line 209: Should be, California, USA.

Response 9. Corrected.

Round 2

Reviewer 3 Report

The authors have answered my queries/comments in their revised manuscript.

Reviewer 4 Report

The authors addressed my concerns